# A Highly Efficient Heterogeneous Catalyst of Bimetal-Organic Frameworks for the Epoxidation of Olefin with H_2_O_2_

**DOI:** 10.3390/molecules25102389

**Published:** 2020-05-21

**Authors:** Fei Wang, Xiang-Guang Meng, Yan-Yan Wu, Hong Huang, Jing Lv, Wen-Wang Yu

**Affiliations:** Key Laboratory of Green Chemistry and Technology, College of Chemistry, Sichuan University, Chengdu 610064, China; wangfei113322@126.com (F.W.); wuyanyan@stu.scu.edu.cn (Y.-Y.W.); 17844622232@163.com (H.H.); lvjing945@163.com (J.L.); yuwenwang@stu.scu.edu.cn (W.-W.Y.)

**Keywords:** metal-organic frameworks, MOF, olefin, epoxidation, H_2_O_2_, catalysts

## Abstract

A series of bimetel organic framework Mn_x_Cu_1−x_-MOF were prepared. The MOFs was characterized and analyzed by powder X-ray diffraction (PXRD), X-ray photoelectron spectroscopy (XPS), scanning electron microscopy (SEM), and transmission electron microscopy (TEM). The catalytic activity of the developed catalyst was tested on various olefins by H_2_O_2_ as oxidant. The MOFs catalyst exhibits excellent catalytic activity for the epoxidations of various aromatic and cyclic olefins. Particularly, Mn_0.1_Cu_0.9_-MOF can achieve 90.2% conversion of styrene with 94.3% selectivity of styrene oxide at 0 °C after reaction 6 h. The MOF exhibited the catalytic activity of inverse temperature effect on epoxidation of styrene. The introduction of copper component can stabilize H_2_O_2_ and inhibit its decomposition to a certain extent. The catalyst can be reused at least five cycles without significant loss in activity towards epoxidation.

## 1. Introduction

The catalytic epoxidation of alkenes play an important role in the organic synthetic chemistry and industrial production, since epoxy compounds are also widely used in synthetic resins, adhesives, coatings, and pharmaceutical synthesis. Homogeneous catalysis with metal complexes is generally highly active for the oxidation of a wide range of olefins [1,2,3]. Recently, iminium salt [4] and ammonium salt [5] were reported as metal-free catalysts for enantioselective asymmetric epoxidation of olefins. As a result of the advantages in catalyst separation and recovery, heterogeneous catalysis systems attracted great attention. Some metal complexes immobilized on solid polymer, grapheme [6,7] or inorganic particles [8,9] were reported as reusable catalysts. Among these heterogeneous catalysts, metal-organic framework materials (MOFs) have received growing interest due to their high BET surfaces, defined pores, ordered structures and adjustable catalytic activity sites of metal ions-organic ligands [10,11]. Maksimchuk et al. used a MIL-125 (Ti) to catalyze the oxidation of cyclohexane by H_2_O_2_, and obtained 41% conversion of cyclohexane with 40% selectivity of cyclohexene oxide [12]. Brown et al. used MOF-525-Mn to catalyze the oxidation of styrene by O_2_, and 99% conversion of styrene with 83.6% selectivity of styrene oxide was observed [13]. Zr and Co based MOFs were also reported as efficient catalysts for the epoxidation of styrene by oxidant TBHP [14,15]. In recent investigations, TBHP, *m*-CPBA, PhIO, and NaIO_4_ were commonly used as efficient oxidant for the epoxidation of olefins [16]. However these oxidants will produce undesired organic wastes [17]. Based on environmental and economic considerations, hydrogen peroxide as a green oxidant has attracted increasingly more attention. Sen et al. synthesized a 3D metal–organic frameworks [Co(H_2_-DHBDA)(bpe)]_n_ to catalyze the oxidation of styrene by H_2_O_2_ and 75% conversion of styrene with 63% selectivity of styrene oxide were obtained [18]. Haddadi et al. prepared PW-MOF by doping W into Cu-framework MOF to catalyze the epoxidation of styrene and achieved 73% conversion of styrene with 83% selectivity of styrene oxide [15]. Molybdenum (VI) modified Zr-MOF catalysts can greatly promote the catalytic epoxidation of cis-cyclooctene, and 88% conversion with 99% selectivity were reported [19]. In addition to metal ion, the binding ligand also plays a key role in the catalytic activity of the reaction. Bagherzadeh et al. used ligand 1,4-benzenedicarboxylic acid (H_2_BDC) to prepare a Co-MOF catalyst to catalyze the oxidation of styrene by TBHP, and obtained 96% conversion of styrene with 45% selectivity of styrene oxide [20]. Wang introduced an additional ligand to the Co-BDC system to prepare a Co-MOF to catalyze the oxidation of styrene by TBHP, and obtained 99% conversion of styrene with 61% selectivity of styrene oxide [21]. Recently, Hu et al. used 2,5-dihydroxyterephthalic acid as ligand to synthesize a bimetal Mn_x_N_100−x_-MOF-74 to catalyze the oxidation of styrene by H_2_O_2_, and 42.5% conversion with 94.9% selectivity of epoxide were achieved [22]. Despite these advances in MOFs catalysis, the epoxidation of olefins, especially aromatic olefins, by hydrogen peroxide is still a challenging task due to its low efficiency and selectivity, or the serious decomposition of hydrogen peroxide by MOFs materials. In recent studies, we found that an iron (II) complex binding with ligand 2-picolinic acid exhibited excellent catalytic activity for epoxidation of terminal olefins with H_2_O_2_ [23]. Herein, we introduced the ligand 2-picolinic acid into metal organic frameworks to prepare a series of bimetal Mn_x_Cu_1−x_-MOFs with two ligands. The MOFs catalysts displayed good activity and selectivity for epoxidation of aromatic and cyclic olefins with H_2_O_2_.

## 2. Results

### 2.1. Characterization of Sample

The synthesized Mn_0.1_Cu_0.9_-MOF was characterized by scanning electron microscopy (SEM) and Transmission electron microscopy (TEM). From the SEM (Figure 1a) and TEM (Figure 1b), it can be seen that the MOFs is sheet-like nanomaterial with layered structure. There are many regular pores about 100–200 nm in diameter. In addition, there are a few small round particles attached to the surface with a diameter of about 40 nm.

Inductively coupled plasma (ICP), Thermo Fisher Scientific. The contents of Mn and Cu in Mn_0.1_Cu_0.9_-MOF were determined by inductively coupled plasma (ICP) analysis. The measurement disclosed the percentage content of Mn was 4.38%, and that of Cu was 51.5%. The value of 0.09 of the molar ratio of Mn to Cu is slightly different from that of the raw material ratio (0.11) of manganese salt to copper salt added in synthesis.

Powder X-ray diffraction (XRD) data were obtained using a Shimazu XRD-6100 diffractometer with Cu-Kα radiation (λ = 1.5406 Å) at 40 kV and 30 mA. The crystallinity of four Mn_x_Cu_1−x_-MOF materials were confirmed by powder X-ray diffraction (PXRD) analysis, the results were illustrated in Figure 2. From Figure 2, it can be seen that the PXRD spectra are similar. Two peaks with 2θ values at 6.6° and 14.5° can be clearly observed.

Fourier transform infrared (FT-IR) spectra were recorded on a Bruker Alpha spectrometer. The FT-IR spectrum of Mn_0.1_Cu_0.9_-MOF was also detected as shown in Figure 3. It can be found that the C=C bond stretching peak of the benzenering appeared in 1542 cm^−1^. The weak peak at 1656 cm^−1^ is related to C=O vibrations. The stretching vibration derived from 2,5-dihydroxyterephthalic acid and 2-picolinic acid. The weak peak of 1244 cm^−1^ is stretching vibration of C-O of 2,5-dihydroxyterephthalic acid. The stretching vibration of C=N in 2-picolinic acid appears at 1470 cm^−1^.

X-ray photoelectron spectra (XPS) were recorded by using an ESCALab220i-XL XPS system. The X-ray photoelectron spectroscopy (XPS) spectrum of elements in Mn_0.1_Cu_0.9_-MOF is shown in Figure 4. From Figure 4 it can be observed that the characteristic peak of Mn2p3/2 appeared at 642.7 ev, which indicates the existence of Mn^2+^. Similarly, the characteristic peak of Cu2p3/2 appeared at 932.7 ev, indicating the existence of Cu^2+^. In the XPS spectrum, we also observed the peaks of C, O, and N.

### 2.2. Catalytic Performances

In previous works, many researchers found that manganese ion and it complexes exhibited good catalytic activity for oxidation of olefins [24], so did Mn-MOFs [25,26]. However, manganese ions and it complexes could decompose rapidly H_2_O_2_ and result in low conversion of olefins. In order to find a suitable target between the decomposition of H_2_O_2_ and the catalytic activity of catalysts, here we synthesized a series of bimetal MOFs materials with various molar ratios of Mn. The oxidations of styrene by H_2_O_2_ catalyzed by these MOFs were carried out. The catalytic activities of Cu-MOF, Mn-MOF, and Mn_0.1_Cu_0.9_-MOF for the oxidation of styrene by H_2_O_2_ are list in Table 1. From Table 1, it can be seen that Cu-MOF showed very low activity, while Mn-MOF and Mn_0.1_Cu_0.9_-MOF displayed excellent catalytic activity. The detailed experimental data for the oxidation of styrene by H_2_O_2_ catalyzed by various Mn_x_Cu_1−x_-MOF are illustrated in Figure 5a. From Figure 5, we can see that the Cu-MOF showed little catalytic activity on the oxidation of styrene. The introduction of a small amount of Mn dramatically improved the catalytic activity of the catalyst. It is observed that the conversion of styrene and the yield of styrene oxide increased rapidly with increasing proportion of Mn. They reached a maximum peak at the mole ratio of 0.1 of Mn to total metal, and decreased slightly and then became stable with increasing content of Mn. The values of conversion of styrene and yield of styrene oxide were 94.3% and 90.2%, respectively, at the ratio of 0.1 of Mn to total metal at 0 °C and after 6 h reaction. In order to understand the catalytic efficiency caused by manganese content, a turnover frequency (TOF) was further calculated as shown in Figure 5b. From Figure 5 it can be obviously observed that the maximum value of TOF was 18 h^−1^ at the mole ratio of 0.1 of Mn to total metal. In addition, we also carried out the experiment of low conversion rate in 3h, and the conversion rate and TOF value were similar to 6h (Appendix A). The addition of more manganese does not improve the activity of the catalyst.

Reaction conditions: 6 mg catalyst, 1 mmol styrene, 0.06 mmol NaHCO_3_, 6 mmol H_2_O_2_, 2 mL DMF, 0 °C, 6 h. Conversion and selectivity were determined by GC using an internal standard method.

### 2.3. Reaction Kinetics

In order to investigate the reaction process, the epoxidation of styrene by H_2_O_2_ catalyzed by Mn_0.1_Cu_0.9_-MOF was carried out. The variety of conversion of styrene and the yield of styrene oxide with reaction time and kinetic curve are shown in Figure 6. From Figure 6, it can be found that although the concentration of H_2_O_2_ was excessive, the reaction kinetic behavior was not consistent with the characteristics of the first-order reaction. At the beginning of the reaction, the conversion rate of styrene was slow and then accelerated. This implied that there may be a binding step between catalyst and substrate in the reaction process. The kinetic behavior provides indirect evidence for the investigation of catalysis mechanism. From Figure 6 it was also observed that the selectivity of styrene oxide remained good throughout reaction process. The yield rapidly reached 75.6% in 2 h, then slowly increased to 85.1% in 2–6 h, and then decreased slightly.

### 2.4. Effect of Reaction Temperature

This catalytic reaction is very sensitive to temperature. The effect of temperature on conversion of styrene catalyzed by Mn_0.1_Cu_0.9_-MOF is shown in Figure 7. Unexpectedly, the conversion showed an inverse temperature effect in the range of 20 °C–35 °C. The conversion of styrene decreased with the increase of reaction temperature. The temperature increases from 20 °C to 35 °C, the conversion of styrene decreases from 66% to 36.5% after 2 h. The phenomenon may be relative to the formation of active oxo species. Under this work’s conditions, peroxymonocarbonate ion (${\mathrm{HCO}}_{4}^{-}$) could form by a labile preequilibrium reaction between bicarbonate ion and H_2_O_2_ (Equation (1)).
(1)$${\mathrm{HCO}}_{3}^{-}+H_{2}O_{2}⇌{\mathrm{HCO}}_{4}^{-}+H_{2}O$$

Kinetic and thermodynamic investigations of Equation (1) give a value of E^0^ (${\mathrm{HCO}}_{4}^{-}$/${\mathrm{HCO}}_{3}^{-}$) = 1.8 V (vs. NHE) [27]. Richardson investigated the formation of ${\mathrm{HCO}}_{4}^{-}$ by Variable temperature ^13^C-NMR spectra and found that in the temperature range of 20 to 40 °C the higher the temperature was, the weaker the ^13^C-NMR spectra signal was [27]. Therefore, it was reasonable to believe that ${\mathrm{HCO}}_{4}^{-}$ was easier to form and participate in the epoxidation of olefins at lower temperature, which leaded to the observation of the reverse temperature effect. At 0 °C, the initial reaction rate was lower than that at other temperatures, but the conversion and epoxy yield were higher after 2 h.

### 2.5. Effect of Solvent and Oxidant

Solvent and oxidant exhibit crucial influence on many oxidations of olefin [28]. In this work, we selected styrene as the substrate to carry out the oxidation reaction by using H_2_O_2_ as oxidant in seven different solvents: 1,2-dichloroethane, toluene, ethanol, tetrahydrofuran, acetonitrile, 1,4-dioxane and *N*,*N*-dimethylformamide, respectively. It can be observed that no reaction could be detected for the oxidation of styrene when 1,2-DCE and Toluene were used as solvent (Table 2, entries 1–2). While, 1,4-Dioxane showed relatively good solvent effect (Table 2, entry 6). *N,N*-dimethylformamide was the excellent solvent system to generate the desired product styrene oxide with 94.3% selectivity and 90.2% conversion at reaction 6 h (Table 2, entry 7).

For the sake of investigate the effect of various oxidants on the epoxidation of styrene, three kinds of oxidants TBHP, PhIO and *m*-CPBA were chosen to carry on the oxidation of styrene. It was observed that the oxidation of styrene didn’t occur when TBHP and PhIO were used as oxidant, respectively (Table 2, entries 8–9). The green oxidant H_2_O_2_ showed the optimal conversion and selectivity of styrene oxide. Further, it was noted that *m*-CPBA displayed only 6.3% oxidation on styrene, which indicated that the prepared MOFs catalyzedepoxidation reaction may not involve peroxy acid catalysis mechanism.

Reaction conditions: 6 mg catalyst, 1 mmol styrene, 0.06 mmol NaHCO_3_, 6 mmol H_2_O_2_, 2 mL DMF, 0 °C, 6 h. Conversion and selectivity were determined by GC using an internal standard method.

### 2.6. Epoxidation of Various Alkenes

Under the optimum reaction conditions, the catalytic oxidations of different substituted aromatic olefins and cyclic olefin were examined. The results were listed in Table 3. For the aryl-substituted terminal olefins 1a and 1b, the substituted group demonstrated different influence on the oxidation reaction. Electron-deficient styrene derivatives 1b gave the corresponding epoxide product along with slower reaction rate (conversion = 88.5%) and slightly lower selectivity (90.2%) (Table 3, entry 2). In particular, this catalytic system is also suitable for the epoxidation of cyclic olefin such as cyclohexene (1d) and cyclopentene (1e) (Table 3, entries 4 and 5).

Reaction conditions: 6 mg catalyst, 1 mmol substrate, 0.06 mmol NaHCO_3_, 6 mmol H_2_O_2_, 2 mL DMF, 0 °C, 6 h. Conversion and selectivity were determined by GC using an internal standard method.

### 2.7. Possible Mechanism of Epoxidation of Styrene

According to our experimental work and relevant data, we propose a possible catalytic mechanism as illustrated in Scheme 1: The calculation shows that C_1_ of terminal olefin is negatively charged (Appendix A), so olefin can replace the carboxyl group on PCA, and combine with Mn^2+^ on bimetal organic frameworks to form complex **A**, then ${\mathrm{HCO}}_{4}^{-}$ which comes from the reaction of H_2_O_2_ with bicarbonate ion, combines with PCA and Mn^2+^ to form complex **B**. The ${\mathrm{HCO}}_{4}^{-}$ can attack positive $C_{2}^{\delta +}$, and the active oxygen caused by heterolysis cleavage of peroxy enters into olefin and then rearrange to form epoxides. After epoxides leaves from complex **C**, The MOFs combine with bicarbonate ion to form complex **D**. Subsequently, the catalyst MOFs recovers by leaving the carbonate ions.

### 2.8. Recycling of Catalyst

The recyclability of catalyst is a considerable aspect of its catalytic ability. Here the reusability of the Mn_0.1_Cu_0.9_-MOF was investigated in the epoxidation of styrene for a reaction time of 6 h in DMF at 0 °C. As depicted in Figure 8, no obvious decrease in catalytic activity was observed after five successive reaction runs. The conversion of styrene reached 87.8% and the selectivity of styrene oxide kept still 86.4%. This suggests that this MOF exhibits excellent catalytic stability and recyclability.

### 2.9. Decomposition of H_2_O_2_

H_2_O_2_ is usually stable in neutral solution. However, metal ion and metal complex can promote the decomposition of H_2_O_2_ [29]. Here we investigated the influence of the prepared MOFs on decomposition of H_2_O_2_, as illustrated in Figure 9. From Figure 9 it can be seen that the Cu-MOF and Mn_0.1_Cu_0.9_-MOF displayed weak activity for the decomposition of H_2_O_2_. The decompositions of H_2_O_2_ catalyzed by Cu-MOF and Mn_0.1_Cu_0.9_-MOF were 3.1% and 4.0% after reacting 6 h, respectively, at 0 °C. Different from Cu-MOF and Mn_0.1_Cu_0.9_-MOF, Mn-MOF showed stronger catalytic activity for decomposition of H_2_O_2_. The decomposition ofH_2_O_2_catalyzed by Mn-MOF was 10.7% after reacting 6 h at 0 °C. The decompositions of H_2_O_2_ catalyzed by Cu-MOF, Mn_0.1_Cu_0.9_-MOF and Mn-MOF were 13.8%, 19.1%, and 69.3% after reacting 6 h, respectively, at 25 °C. The strong decomposition of manganese to hydrogen peroxide is disadvantageous to the epoxidation of olefins. The introduction of a small amount of manganese ions into the Cu-MOF not only improves the catalytic activity of MOFs, but also solves the problem of strong decomposition of H_2_O_2_.

## 3. Materials and Methods

### 3.1. Materials

Manganese(II) chloride tetrahydrate, cupric nitrate trihydrate, sodium bicarbonate, styrene, hydrogen peroxide, *N*,*N*-Dimethyl formamide (DMF), acetonitrile, 1,4-Dioxane, tetrahydrofuran (THF), 1,2-dichlorethane (1,2-DCE), toluene, ethanol, and methanol were purchased from Jinshan Chemical Company(Chengdu, China), 2,5-dihydroxyterephthalic acid (H_4_DHTA), 2-picolinic acid (PCA), *m*-chloroperoxybenzoic acid (*m*-CPBA), *tert*-butyl hydroperoxide (TBHP), and iodosylbenzene (PhIO) were purchased from Adamas Regent Company. All the chemicals were analytical grade and purchased from commercial sources and used without further purification.

### 3.2. Preparations of Mn-MOF, Cu-MOF, and Mn_x_Cu_y_-MOF

The Mn-MOF was synthesized as follows: MnCl_2_·4H_2_O (0.757 mmol), H_4_DHTA (0.126 mmol) and PCA (0.126 mmol) were dissolved in 17 mL DMF containing 1 mL of ethanol and 1 mL of H_2_O. Subsequently, the solution was moved into 25 mL teflon-lined autoclave and heated at 120 °C for 24 h. After cooling to room temperature, the product was collected by centrifugation and washed three times with DMF and CH_3_OH, respectively. Finally, the crystallized Mn-MOF of 119 mg was obtained through drying in a vacuum for 12 h. By the similar procedures, the Cu-MOF and Mn_x_Cu_1−x_-MOF with various molar ratio of Mn to total metal were obtained.

### 3.3. Epoxidation of Olefin

The typical catalytic reaction was carried out as follows: 1 mmol styrene and 6 mg MOFs were added to a 25 mL round bottom flask containing 2 mL DMF. Then, the solution was dripped into 0.9 M 67 μL NaHCO_3_ aqueous solution containing 612 μL H_2_O_2_ of 30% (*w*/*w*). The solution was kept at 0 °C in an ice bath and stirred continuously for 6 h. The concentrations of reaction substrates and products were determined by GC through the comparison of the peak area of analyte with that of internal standard.

## 4. Conclusions

In this work, we prepared a series of bimetallic organic framework Mn_x_Cu_1−x_-MOF with two ligands by hydrothermal synthesis. The MOFs was characterized and analyzed by PXRD, XPS, SEM, and TEM. The Cu–MOF showed low catalytic activity for both epoxidation of olefins and decomposition of H_2_O_2_, while Mn-MOF showed good catalytic activity for both. The Mn_0.1_Cu_0.9_-MOF exhibits excellent catalytic activity for the epoxidations of various aromatic and cyclic olefins and weak activity on decomposition of H_2_O_2_. Styrene can be oxidized by H_2_O_2_ and the yield of styrene oxide achieves 85% in the presence of Mn_0.1_Cu_0.9_-MOF at 0 °C after reaction 6 h. The decomposition of H_2_O_2_ catalyzed by Mn_0.1_Cu_0.9_-MOF was 4.0% after reacting 6 h at 0 °C. The inverse temperature effect in catalytic epoxidation reaction was discussed. A mechanism of peroxybicarbonate-assisted catalysis was suggested. The catalyst can be reused at least five cycles without significant loss in activity towards epoxidation. The conversion of styrene reached 87.8% and the selectivity of styrene oxide kept still 86.4% after five cycles.

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
