# Peer review of "A Highly Efficient Heterogeneous Catalyst of Bimetal-Organic Frameworks for the Epoxidation of Olefin with H_2_O_2"

_molecules, 2020, doi:10.3390/molecules25102389_

Round 1
Reviewer 1 Report
Finding new efficient solid catalyst to substitute homogeneous catalytic processes by heterogeneous ones for the epoxidation of olefins is still a challenging issue. In this context, this work reports some interesting results but it also presents a series of important weaknesses that should be solved before the work can be published.
1) The authors propose to introduce ligand 2-picolinic acid together with various Mn contents into metal organic frameworks containing also Cu (based on a previous work using Fe instead of Mn). Nevertheless, they do not explain clearly enough the reasons of the choice of this system and the specific roles expected from each of the components: notably, which are the catalytic sites? what is the expected effect of the ligand? Why both Mn and Cu? etc.
2) The composition of the MnxCuy-MOF series of samples is never presented. It should be clearly introduced in section 2 (and detailed in section 3.2):
- what is the meaning of x and y (atom %? weight %, other %)?
- how is the x/y ratio varied?
- is the Cu content constant or does it vary as well?
3) The caption of Figure 1 is wrong and should be corrected (together with the related text, line 68): a) is for SEM and B) for TEM.
4) The authors conclude from Fig. 2a that a small amount of Mn dramatically improves the catalytic activity (this is true), but an almost full conversion is then reached (close to 90%), so it is probably not correct to say that “adding more manganese does not improve further the activity”. Indeed, once almost all reactant has been converted (by the low Mn amount corresponding to x=0,1), there is no reactant left to be transformed (and many Mn “active” sites stay inactive). To be relevant, the comparisons should be done at lower conversion levels (using for instance lower amounts of catalysts).
5) The calculation of turnover over frequencies requires to know the number of active sites. How were the TOF values established to plot Figure 2b (Cu? Mn? Both?)? Such issue might be especially complex for a heterogeneous catalyst that is a solid in which only a part of the atoms is usually accessible to reactants (surface atoms). Moreover, the comparisons need to be done far from full conversions (same as in comment 4), otherwise part of the active sites might be not working (making the TOF values meaningless).
6) Explain better the comments on kinetics in section 2.3: excess of H2O2? what shape of curve is expected with a first order reaction? what is meant by a “binding step”? what is the “evidenced mechanism” (line 113)?
7) The effect of the nature of the solvent shown in Table 2 is observed but without any attempt of explanation. Why are the 1,2-DCE and Toluene bad solvents and 1,4-Dioxane and N,N-dimethylformamide good ones?
8) The mechanism proposed in section 2.6 looks like a nice output of the work but its links with the experimental data are confuse. More explanations based on the observations should be given: what are the data used? Where are the evidences? etc.
9) The text contains many typing errors, as exemplified as follows (corrected sentences): in recent investigations (Line 40); can greatly promote (line 48); In recent studies (line 60); by scanning electron microscopy (line 67); by H2O2 catalyzed (a space missing, line 82); From this Table, is can be seen that (line 84); are illustrated (line 86); are listed in Table 1 (line 83); It is observed that (line 89); are shown in Figure.3. From this Figure (line 108); From Figure.3 (line 113); is shown in Figure. 4 (line 120); that the prepared MOFs catalyzed epoxidation (a space missing, line 150); The solution was moved (line 216); the product was collected (line 217); 2 ml DMF. Then (a sapce missing, line 223).
10) Some sentences or expressions are wrong (should be corrected): the catalytic activity of the reaction (line 80); with various ratios of Mn to Mn and Cu (lines 81, 91, 93, 96).
Reviewer 2 Report
The manuscript “A highly efficient heterogeneous catalyst of bimetal-organic frameworks for the epoxidation of olefin with H2O2” by Wang et al. describes the use of Mn/Cu-MOF as a heterogeneous catalyst for olefin epoxidation. The experimental work seems reliable, but the presentation does not meet the criteria for the publication in Molecules. I recommend that the authors revise their work substantially prior to the possible re-submitting in any journal.
The language should be polished, as the text is full of typos, missing spaces etc. and therefore it is tedious to read.
MOFs are made by mixing Mn and Cu precursors in different ratios with organic acids. Are there any elemental analyses or other characterisations to prove the ratio of the metal ions in the final product? Yields should be given in the experimental part – even if the reactions were quantitative.
In 2.1. Characterization of sample, it is said” the MOFs is nanorod with layered structure” and there are “pores about 100-200nm” as well as “small round particles attached to the surface with a diameter of about 40 nm”. However, I cannot see any nanorods in the SEM image but more like sheets or flakes. Pores and round particles are seen in TEM image, not in SEM.
2.2. Catalytic performance: I do not necessarily understand the expression “various ratios of Mn to Mn and Cu”.
References must be reformulated. Reference 2, for instance, should be as M. Mitra, O. Cusso,… E. Nordlander, (i.e. given name abbreviated, family name shown).
Finally, I do not believe the given reaction mechanism where terminal alkene can substitute a chelating carboxylate. The authors tell that it is “according to experimental work and relevant data” and later say “the calculation shows that C1 of terminal olefin is negatively charged..”. I cannot see any connection with the given experimental data and the reaction mechanism. Moreover, where is the calculation shown? Do the authors refer to some previous work or should it be given here?
Reviewer 3 Report
Referee of the paper “A highly efficient heterogeneous catalyst of 3 bimetal-organic frameworks for the epoxidation of 4 olefin with H2O2”
Journal: Molecules
Manuscript ID: molecules-660207
Article Type: Paper
Date Submitted by the
Author: 9-Dec-2019
Commentary to the authors: The article reports a series of bimetel organic framework MnxCu1-x-MOF and the morphology of MOFs was characterized by scanning electron mocroscopy (SEM) and transmission electron microscopy (TEM). It the manuscript it was showed the catalytic activity of the developed catalyst on various olefins by H2O2 as oxidant. The MOFs catalyst exhibits high catalytic activity for the epoxidations of various aromatic and cyclic olefins. Specially, Mn0.1Cu0.9-MOF can achieve 90.2% conversion of styrene with 94.3% selectivity of styrene oxide at 0 °C after reaction 6 h. The MOF exhibited the catalytic activity of inverse temperature effect on epoxidation of styrene. It is noteworthy that the introduction of copper component can stabilize H2O2 and inhibit its decomposition to a certain extent. The catalyst can be reused at least five cycles without significant loss in activity towards epoxidation. The obtained scientific data are interested, and the reported literature is well defined and justified for all mentioned data in the introduction. However, it is noted some English mistakes and it is important to fix, for example, the following items:
In the abstract (line 17): the sentence “at 0°Cafter reaction 6 h” needs to be corrected to “at 0 °C after reaction 6 h”. See other sentences that have the same mistake in the manuscript.
Please take care in all document and correct all spaces between words as well as spaces between numbers and units. Correct units. Subscript and superscript numbers. Include what means each abbreviation.
In page 3 (lines 84 and 85): The authors refer "From Table 1 we found that Cu-MOF showed very low activity, while Mn-MOF and Mn0.1Cu0.9-MOF displayed excellent catalytic activity”. Do you have a possible explanation for that?
The conclusion can be improved with all supported data that were obtained in the experimental measurements for the manuscript.
Round 2
Reviewer 1 Report
I am not fully convinced by some of the answers (in particular that on TOFs estimation) and I still find part of the explanations or interpretations of fair-medium quality. Nevertheless, the revised version is significantly improved compared to the initial document (and some relevant data have been added) and it presents a certain number of results that could be of some interest for people working in the domain. Hence, I consider that the work can be published after an additional careful reading to (at least!) remove the remaining (and still numerous) typing or sentences errors, such as: bimetal (line 15); “for various olefins epoxidation using H2O2 as oxidant” (line 18); plays (line 29); by scanning (without the) (line 72); From the SEM (Figure. 1 (a)) and TEM (Figure 1 (b)) images of Mn0.1Cu0.9-MOF…(line 73); check line 83 (not a sentence); disclosed that (line 85); was confirmed (line 90); are illustrated (line 90); The Fourier transform infrared (FT-IR) spectrum of Mn0.1Cu0.9-MOF, recorded on a Bruker Alpha spectrometer, is shown in Figure 3 (lines 104-105); check the sentence lines 107-108 (the verb is missing); its complexes (line 138), exhibit (line 138); its complexes (line 140); decompose rapidly H2O2 and result (instead of could decompose.., line 140); The oxidation (without s, line 143); was (line 143); are listed (line 145); etc. (I have stopped checking typing errors at this point, please pay great attention to the following). Also, the conversion and selectivity values given in Table 1 are presented with a precision that is totally meaningless considering the margin errors existing on the instruments used for such measurements: at least the second numbers after commas (if not the first ones!) should be systematically removed.
Author Response
We understand what you said the comparisons need to be done far from full conversions. When the reaction time is 3h, the conversion of styrene is less than 60%. We think that all manganese on the surface of catalyst is still working at that moment. The plots of conversion of styrene and the yield of styrene oxide and the TOF versus a series of MnxCuy-MOFs for reaction 3h were added in the SI Figure S2. From Fig.S2 it can be found that the plots of the TOF versus a series of MnxCu1-x-MOFs showed similar feature to Fig.5b in revised version.
We appreciate the reviewer for the carefully check for our manuscript. According to your suggestions, we have corrected the errors of typing or sentences in the manuscript.
Thank your careful examination for our manuscript.
Reviewer 2 Report
The revised version is significantly improved compared to the first version and it may be published after some language editing. In the discussions on the catalytic cycle, the authors should refere the calcalations presented in the supplementary material instead of a laconic statement "The calculation shows that C1 of terminal olefin is negatively charged".
Author Response
We appreciate the suggestions of reviewer. In order to understand the electrostatic induction effect, we calculated the NBO charge distribution of atoms on styrene by the density functional theory (DFT) method at the B3LYP/ (6-311G (d) level using Gaussian 09 program. The calculated results were shown in Figure S1. The discussion was added in the revised paper.